# A metagenomics survey of viral diversity in mosquito vectors allows the first detection of Sindbis virus in Burkina Faso

Didier P. Alexandre Kaboré[1,2], Antoni Exbrayat[3], Floriant Charriat[3], Dieudonné Diloma Soma[1,2], Simon P. Sawadogo[1], Georges Anicet Ouédraogo[2], Edouard Tuaillon[4,5], Philippe Van de Perre[4,5], Thierry Baldet[3], Côme Morel[3], Roch K. Dabiré[1]*, Patricia Gil[3]*, Serafin Gutierrez [3,5]*

1 Institut de Recherche en Sciences de la Santé (IRSS), Bobo-Dioulasso, Burkina Faso, 2 Université Nazi BONI, Bobo-Dioulasso, Burkina Faso, 3 ASTRE Research Unit, CIRAD, INRAe, Montpellier University, Montpellier, France, 4 Pathogenesis and Control of Chronic and Emerging Infections, INSERM, University of Montpellier, Établissement Français du Sang, Montpellier, France, 5 Laboratory of Virology, Montpellier University Hospital, Montpellier, France

* dabireroch@gmail.com (RKD); patricia.gil@cirad.fr (PG); serafin.gutierrez@cirad.fr (SG)

## Abstract

Arboviruses (i.e., Arthropod-borne viruses) pose a threat to human health worldwide. This taxonomically-diverse group includes numerous viruses that recurrently spread into new regions. Therefore, periodic surveys of the arboviral diversity in a given region can help optimize the diagnosis of arboviral infections. However, such surveys are infrequent, especially in low-income countries. Consequently, case investigation is often limited to a fraction of the arboviral diversity. This situation is likely to result in undiagnosed cases. Here, we investigated the diversity of mosquito-borne arboviruses in two regions of Burkina Faso. To this end, we used untargeted metagenomics to screen mosquitoes collected over three years in six urban and rural areas. The analysis focused on two mosquito species, *Aedes aegypti* and *Culex quinquefasciatus*, considered to be among the most important vectors of arboviruses worldwide. The screening detected Sindbis virus (SINV, *Togaviridae*) for the first time in Burkina Faso. This zoonotic arbovirus has spread from Africa to Europe. SINV causes periodic outbreaks in Europe but its distribution and epidemiology in Africa remains largely unstudied. SINV was detected in one of the six areas, and at a single year. Detection was validated with isolation in cell culture. SINV was only detected in *Cx. quinquefasciatus*, adding to the list of potential vectors of SINV in nature. The SINV infection rate in mosquitoes was similar to those observed in European regions experiencing SINV outbreaks. Phylogenetic analysis placed the nearly-full genome within a cluster of Central African strains of lineage I. This cluster is thought to be at the origin of the SINV strains introduced into Europe. Our results call for studies on the prevalence of SINV infections in the region to estimate the disease burden and the interest of SINV diagnostic in case investigation.

**Data availability statement:** All relevant data are within the paper and its Supporting Information files.

**Funding:** This work was supported by the Montpellier University of Excellence programme, 2018 (ArboSud project; authors receiving the grant: P.V.T., R.D., T.B., S.G.). D.A.P.K. was the recipient of a funding by European Union's Horizon 2020 research and innovation programme (Infravec2 project, grant agreement 731060). The funders had no role in study design, data collection and analysis, decision to publish, or preparation of the manuscript. There was no additional external funding received for this study.

**Competing interests:** The authors have declared that no competing interests exist.

## Introduction

Emergence and re-emergence of arboviral diseases have become more frequent and intense over the last decades [1]. This trend is likely due to a combination of factors, including global transport, urbanization and climate change [1,2]. The increase in incidence and geographic distribution has been observed for different virus species representative of the large taxonomic diversity of arboviruses [3,4]. To date, this diversity potentially includes 245 virus species of RNA viruses, with most species belonging to the families *Flaviviridae* and *Togaviridae*, and the order *Bunyavirales* [5]. This diversity includes viruses with different life cycles (*e.g.*, differing in the mosquito and vertebrate species they infect) [2]. Not surprisingly, these viruses are associated with different epidemiological situations. For example, dengue virus (DENV) stands out in terms of case counts, with hundreds of thousands of cases per year worldwide [6]. On the other hand, numerous lesser-known viruses often cause outbreaks that are limited in case count, geographic distribution, or duration [7].

There is currently no vaccine or antiviral drug available for most arboviruses [8–10]. Prevention and control of arboviruses therefore rely mainly on surveillance and vector control. When viruses are detected, mosquito management and public awareness campaigns are usually implemented to reduce exposure to infected vectors [11]. However, the surveillance of these viruses often faces several challenges. One of these challenges is the large diversity of arbovirus species harbored in certain regions [7]. For example, several tens of arbovirus species can be found in several African countries [7]. This diversity complicates the implementation of diagnostic tests for all the viruses present in the region, primarily due to test availability and cost [12]. Another challenge is the change in the diversity of arboviruses in a given region over time. Many of these viruses have periodically spread over long distances and colonized new areas in which populations of mosquito vectors exist. Such long-distance spread is mainly driven by infected vertebrate hosts, such as humans or migratory birds [13–17]. When virus spread to non-endemic regions results in outbreaks, these viruses may go undetected because infections often result in non-specific signs and symptoms.

Thus, the diversity of arboviruses transmitted by mosquitoes can be high in certain regions and can change over time. However, routine diagnosis of potential infections is usually limited to a few arboviruses with a major impact on public health, like DENV, even in countries where arboviral diversity is high [18]. This situation can lead to diagnostic failure. Although specific treatments for arboviral infections are currently unavailable, early diagnosis can prevent unspecific diagnostic testing and treatment, and provide diagnostic and prognostic information to patients. In addition, the resulting lack of data on disease prevalence hinders the design and implementation of appropriate surveillance and control strategies. Periodic surveys of arboviral diversity could mitigate these problems by providing updated reports on the viruses actively transmitted in a given region. However, comprehensive surveys of the arboviral diversity are rare, especially in low-income countries [18].

Here, we aimed to characterize the arboviral diversity in two regions of Burkina Faso. This West African country is highly connected to other regions in Africa and

other continents by human and animal movements [19]. Several arboviruses, including DENV, Zika virus (ZIKV), chikungunya virus (CHIKV), West Nile virus (WNV), yellow fever virus and Rift Valley fever virus, have been previously detected in Burkina Faso [20–23]. Nevertheless, studies have focused on a limited number of these viruses. Here, arboviral diversity was analyzed in a mosquito collection, a classic approach in arbovirus surveillance [24]. Two mosquito species were selected, *Aedes aegypti* and *Culex quinquefasciatus*, both considered to be among the most important vectors of arboviruses worldwide. These mosquitoes differ in the diversity of arboviruses they transmit [25]. To further facilitate a comprehensive exploration of viral diversity, environments with different levels of human activity, and thus likely to harbor different mosquito vectors and the viruses they carry [26], were sampled for three consecutive years. We had previously screened this mosquito collection for several arboviruses – including DENV, ZIKV, WNV and CHIKV – using RT-PCR [20,21].

In this study, we have used shotgun metagenomics to screen the mosquito collection. Metagenomics is an untargeted diagnostic approach that allows a comprehensive exploration of the viral diversity in a sample, thus avoiding the use of a multitude of targeted diagnostic tests [27]. Our approach allowed the first detection of Sindbis virus (SINV; family *Togaviridae*, genus *Alphavirus*, species *Alphavirus sindbis*) in Burkina Faso. The cycle of this zoonotic arbovirus mainly involves mosquitoes and wild birds [28]. SINV can be transmitted to humans through mosquito bites. The main symptoms are fever, rash and arthralgia [28,29]. A significant proportion of clinical cases result in joint symptoms for months or years [30]. However, most infections are subclinical, resulting in high rate of undiagnosed infections [28]. To date, periodic outbreaks have been reported only in Finland, Sweden, Russia and South Africa [28,31]. The limited geographic extent of these outbreaks contrasts with the quasi-global distribution of SINV, with reports from Africa, Asia, Australia and Europe [28]. This epidemiological pattern can be partially explained by viral genetics. SINV strains are classified into six genotypes with a limited or no overlap in geographic range [28]. Genotype 1 is the only genotype detected in Europe and Africa, the regions where outbreaks have been reported. This situation thus suggests an association between SINV genotype and infection risk. Moreover, European strains are the result of a few introductions from Africa, probably through bird migration [16]. SINV strains from central Africa have been identified as the potential ancestors of the European strains [16]. However, the geographical origin of the European strains remains to be further investigated because SINV sequences are only available from a limited number of African regions. In addition, recent reports suggest that SINV is either spreading to new African regions or is endemic to a larger area than previously thought [32,33]. Thus, the paucity of data on SINV hampers to fully assess its distribution in Africa and its dissemination routes.

## Methods

### Mosquito sampling

The sampling sites have been previously described [34]. The choice of the sampling sites aimed at exploring the viral diversity in different geographical regions and human-activity levels. Mosquitoes were collected in two regions of Burkina Faso, the Hauts-Bassins and Southwest regions (Fig 1). These regions have an average annual rainfall of 1 200 mm. The climate is tropical with two seasons: a rainy season from June to September and a dry season from October to May. The vegetation consists mainly of tree or wooded savannahs in the Hauts-Bassins and Southwest regions, respectively. In each region, urban and rural zones were sampled. In the Hauts-Bassins region, three zones were sampled, including one urban zone (Urban 1 zone) and two rural zones (Rural 1 and Rural 2 zones). The Urban 1 zone included three samping sites in the city of Bobo-Dioulasso, the second largest city in Burkina Faso (904 920 inhabitants and 13 680 ha). The Rural 1 zone comprised four sites in a rural area dominated by rice fields 30 km to the north of Bobo-Dioulasso. The Rural 2 sites were located in two forested areas situated 18 km far away from Bobo-Dioulasso (Nasso and Dinderesso forests). In the Southwest region, three zones (Urban 2, Urban 3 and Rural 3 zones) were distributed along a transect linking two cities, Diébougou (25 688 inhabitants) and Gaoua (45 284 inhabitants). Urban 2 and Urban 3 zones included a site in either Diébougou or Gaoua, respectively. Moreover, Rural 3 zone included four sites situated in rural areas along the road connecting these two cities. No permits were required for the mosquito collection.

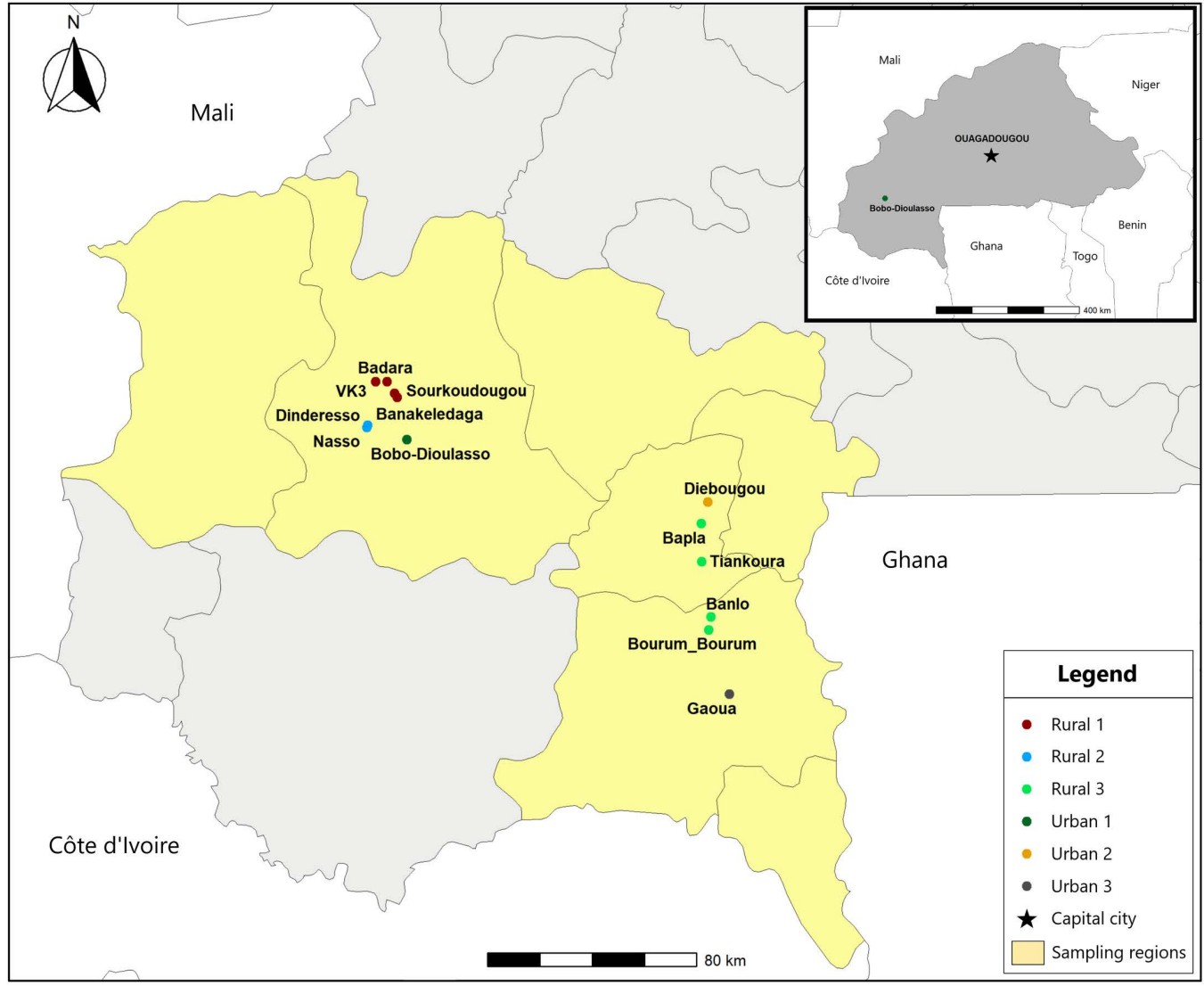

**Fig 1. Mosquito collection sites.** Dots represent collection sites. Dot color stands for sampling zone (rural and urban zones). The inner panel shows the country of Burkina-Faso with the locations of the capital city (Ouagadougou) and the city of Bobo-Dioulasso. Maps were generated with the rnaturalearth package in R and base layers from the Natural Earth dataset (http://www.naturalearthdata.com/). All maps are in the public domain.

Mosquito sampling has been described in detail previously [34]. Briefly, mosquito collection was conducted at all sites during three periods: August/September 2019, June/October 2020 and May/June 2021. The collection of adult mosquitoes was carried out on two consecutive days in each site. Three trapping methods were used in each site: the double net tent [35], the BG-Sentinel traps (Biogents, Germany) and the Prokopack aspirator [36]. Individual mosquitoes were identified morphologically using identification keys [37,38] on an ice-cold platform to avoid viral genome degradation. Mosquito numbers per sampling method have been provided in [34].

Immediately after species identification, non-blood-engorged females were pooled by mosquito species, date and collection zone (Table 1). Pool size was set at 30 individuals but had to be adjusted depending on catch size (median/minimum/maximum pool size = 30/6/37; S1 Table). Pools were immediately stored dry at −80°C for subsequent analyses. A total of 59 pools of *Cx. quinquefasciatus* (1 629 females) and 47 pools of *Ae. aegypti* (1 356 females) were generated

**Table 1. Number of *Aedes aegypti* or *Culex quinquefasciatus* females per collection zone and year, and the resulting pools and libraries for Illumina sequencing. Totals per collection zone are shown in bold. The last row shows totals for all zones combined.**

| zone/year | *Aedes aegypti* | | | *Culex quinquefasciatus* | | |
|---|---|---|---|---|---|---|
| | Females | Pools | Libraries | Females | Pools | Libraries |
| **Rural 1** | **166** | **6** | **6** | **325** | **12** | **7** |
| 2019 | 90 | 3 | 3 | 17 | 1 | 1 |
| 2020 | 51 | 2 | 2 | 172 | 6 | 3 |
| 2021 | 25 | 1 | 1 | 136 | 5 | 3 |
| **Rural 2** | **47** | **2** | **2** | **158** | **6** | **4** |
| 2019 | 0 | 0 | 0 | 0 | 0 | 0 |
| 2020 | 35 | 1 | 1 | 81 | 3 | 2 |
| 2021 | 12 | 1 | 1 | 77 | 3 | 2 |
| **Rural 3** | **79** | **3** | **3** | **101** | **4** | **4** |
| 2019 | 0 | 0 | 0 | 0 | 0 | 0 |
| 2020 | 52 | 2 | 2 | 44 | 2 | 2 |
| 2021 | 27 | 1 | 1 | 57 | 2 | 2 |
| **Urban 1** | **609** | **21** | **11** | **616** | **21** | **10** |
| 2019 | 57 | 2 | 2 | 77 | 3 | 2 |
| 2020 | 420 | 14 | 7 | 397 | 13 | 5 |
| 2021 | 132 | 5 | 2 | 142 | 5 | 3 |
| **Urban 2** | **278** | **9** | **6** | **195** | **7** | **4** |
| 2019 | 0 | 0 | 0 | 0 | 0 | 0 |
| 2020 | 184 | 6 | 3 | 83 | 3 | 2 |
| 2021 | 94 | 3 | 3 | 112 | 4 | 2 |
| **Urban 3** | **177** | **6** | **4** | **234** | **9** | **5** |
| 2019 | 0 | 0 | 0 | 0 | 0 | 0 |
| 2020 | 121 | 4 | 2 | 48 | 2 | 2 |
| 2021 | 56 | 2 | 2 | 186 | 7 | 3 |
| **TOTAL** | **1 356** | **47** | **32** | **1 629** | **59** | **34** |

(Table 1 and S1 Table). In 2019, mosquito catches were low and pools could only be generated for the Rural 1 and Urban 1 zones.

### Isolation of nucleic acids

Each pool was processed to obtain nucleic acids enriched in nuclease-protected molecules as previously described [39]. Briefly, mosquitoes were homogenized in 500 µl ice-cold 1X phosphate buffered saline buffer using two ice-cold steel bearing balls (3 mm diameter, LOUDET) in a TissueLyser II (Qiagen) and clarified by centrifugation. A 150-µl aliquot of clarified homogenates was digested with a nuclease cocktail consisting of 20 U/L of exonuclease I (Thermofisher), 5 U/L of RNase I (Thermofisher), 25 U/L of benzonase (Merck Chemical) and 20 U/L of turbo DNase (Ambion). Then, nucleic acids in the resulting suspension were isolated using the Nucleospin RNA virus kit (Macherey Nagel) according to the manufacturer's protocol with modifications. Specifically, 20 µl of proteinase K at 20 mg/ml (Macherey Nagel) were added per reaction in the RAV1 buffer. Moreover, the RNA carrier in RAV1 buffer was replaced by 5 µg of linear acrylamide (Ambion). Nucleic acids were eluted in 50 µl of RNase-free water and stored at −80°C. The quality and quantity of the RNA was estimated using capillary electrophoresis (2100 Bioanalyzer, Thermofisher). In addition, a second isolation per pool was performed as described above but without the nuclease treatment. The resulting total RNA was used for virus detection by RT-qPCR.

## Library preparation and Illumina sequencing

Custom library preparation for Illumina sequencing was performed from nucleic-acid suspensions enriched for nuclease-protected molecules as described [39]. Suspensions from pools were distributed in libraries to obtain a similar number of libraries per mosquito species, zone and period. Briefly, cDNA was generated in a reverse transcription reaction using the RevertAid First Strand cDNA Synthesis kit (ThermoScientific) and the 454-E-8N primers. Double-stranded DNA (dsDNA) was then generated using the Klenow fragment polymerase (Fisher Scientific) and the 454-E-8N primers. The dsDNA fragments were amplified in a PCR reaction using the 454-E primer and the Phusion High-Fidelity DNA Polymerase kit (Fisher Scientific). Amplicons were purified using the NucleoSpin gel and PCR Clean-up (Macherey-Nagel). Adapters were then ligated to the amplicons in a PCR reaction using the P5 and P7-bearing adapter primers and the Phusion High-Fidelity DNA Polymerase kit (Thermo Scientific). Amplicons were sized and purified using an AMPure XP Magnetic Bead Capture (Agencourt). The size of the resulting libraries (expected 500–600 bp; bead to sample ratio = 0.6) was validated by capillary electrophoresis (Agilent 2100 Bioanalyzer, Agilent Technologies). Library concentration was estimated using the Library Quantification kit (Takara Bio) according to the manufacturer's protocol. All libraries were pooled together in similar concentrations and the resulting sample was sequenced in the same run with a HiSeq 2500 sequencer (Illumina; 250-bp paired reads) using specific sequencing primers [39].

## Virus detection from Illumina reads

Bioinformatic analysis of the reads was carried out using the Snakevir pipeline [39]. Snakevir and its documentation are freely available at https://github.com/FlorianCHA/snakevir. Briefly, adapter and low-quality sequences were removed from reads using Cutadapt 1.6 [40]. Subsequently, rRNA-derived reads were removed from the dataset. Identification of rRNA-derived reads was done by mapping reads with BWA 0.7.15 [41] against rRNA sequences (SILVA bacterial bases: SSURefNr99 and LSURef; 18/01/2017; SILVA dipteran base, release 132) [42]. Then, reads from all libraries were pooled and used as input for a de-novo assembly to generate a non-redundant set of contigs using Megahit v1.1.2 [43]. The contigs and unassembled reads were used for a second *de-novo* assembly with CAP3 [44]. The resulting metagenome was screened for virus-derived contigs in a homology search using Diamond v2.1.8 [45] (e-value cutoff = $10^{-3}$) against the NCBI nr database (May 2021). A further search for SINV-like contigs was performed by assembling the virus-like contigs that mapped to a SINV genome (Accession OK644705) in Geneious 10.2.6.

## PCR detection of SINV and Sanger sequencing

A SINV-specific real-time RT qPCR [32] was used to screen the 60 pools of *Cx. quinquefasciatus* mosquitoes for the presence of SINV nucleic acids. RT-qPCR was performed using the Luna Universal One-Step RT-qPCR Kit (Biolaps) according to the manufacturer's protocol on a LightCycler real-time PCR thermocycler (Roche).

SANGER sequencing was performed on PCR amplicons covering gaps between SINV contigs. Amplicons were obtained using a set of primers (S2 Table), designed on contig sequences, with the Phusion Hight-Fidelity PCR Kit (Thermofisher) according to the manufacturer's instructions.

## Analysis of the infection rate

Infection rate analysis based on PCR data was performed using the *binGroup* package [46] in R software. This package provides functions to estimate the infection rate from the analysis of pools or groups, rather than individuals, for virus infection. Here, estimates and their confidence intervals were obtained using the pooledBin function. This function follows a maximum likelihood approach to calculate confidence intervals for a single infection rate based on tests of pools containing pools of different sizes. We estimated the infection rate in the Rural 1 zone in 2020, the only zone and year in which a SINV-positive library was detected. The dataset and the R script are available in S3 Table and S1 Script. Only the pool that resulted in SINV infection in cell culture was considered as SINV positive.

## Phylogenetic analysis

Multiple alignment of 39 SINV sequences, including the sequence from Burkina Faso, was performed using MEGA version 11 software [47]. The sequences were concatenated sequences of the ORFs of the nonstructural protein (nsP) and the structural protein (sP). Sequences were aligned using the Clustal W multiple alignment algorithm [48]. A phylogenetic tree was constructed using the maximum-likelihood method and a General Time Reversible model [49] with 1 000 bootstrap replicates using MEGA version 11 software.

## Virus isolation

Homogenates of the two mosquito pools in which SINV was detected were used as inoculum for virus isolation in Vero African green monkey cells (Vero ATCC CCL81). For each pool, a volume of 100 µl of homogenate was inoculated into a T-25 cell culture flask and incubated at 37°C in a 5% $CO_2$ atmosphere for one and half hours. The supernatant was then replaced with 5 ml of MEM medium supplemented with 10% FBS, 200mM L-glutamine, 1% penicillin-streptomycin, 50 µg/ml gentamycin. Cells were incubated at 37°C in a 5% $CO_2$ atmosphere and examined daily for cytopathic effects. The supernatant (150 µl) from the first passage (five days after inoculation) was analyzed for the presence of SINV nucleic acids using the SINV RT-qPCR as described previously [32].

## Results

We used shotgun sequencing of nuclease-protected nucleic acids to investigate the viral diversity in populations of *Cx. quinquefasciatus* and *Ae. aegypti* from two regions of Burkina Faso. A total of 66 libraries were generated from 2 985 mosquitoes and sequenced. Differences in the size of the mosquito collections led to differences in the number of libraries per region (40 and 26 libraries from the Hauts-Bassins and Southwest regions, respectively; Table 1). The number of libraries was similar between mosquito species but with differences in the number of mosquitoes per library (median number of mosquitoes per library = 51.5 and 31.5 for *Cx. quinquefasciatus* and *Ae. aegypti,* respectively; Table 1 and S1 Table).

Sequencing generated a total of 216 million reads and an average of 3.34 million reads per library (min/max = 0.8/7.65 million reads per library; SRA data available in GenBank BioProject PRJNA1099472). The total number of viral reads per mosquito species or the mean number of viral reads per female was similar between mosquito species (total reads/mean reads per female = $1.07 \times 10^6$/657 and $1.09 \times 10^6$/804 reads for *Cx. quinquefasciatus* and *Ae. aegypti* respectively). After bioinformatic processing of the reads, we identified 1 412 virus-like contigs (mean/min/max contig length = 1.14/0.22/28.36 Kb). Virus-like contigs were associated with 103 virus species and mostly with mosquito-specific viruses (i.e., viruses that only infect mosquitoes, as opposed to arboviruses that infect both vertebrates and arthropods; 85% of the virus species). Since the goal of this work is to detect arboviruses, the diversity of mosquito-specific viruses is described in a companion manuscript [50] .

A zoonotic arbovirus was detected. This virus was SINV. A total of twelve contigs were found associated with SINV sequences with percent identities at the amino acid level greater than 90%. The SINV-like contigs were found in a single library (library Cx_R1_20_F_2; S1 Table). The library derived from two pools of *Cx. quinquefasciatus* collected in Rural 1 zone in 2020 (S3 Table). The SINV contigs covered 84% of the genome (9 712 bp). Gaps between contigs were determined by Sanger sequencing. A nearly-complete genome (11 244 bp) was obtained and submitted to GenBank (accession OR659083). A phylogenetic analysis clustered the SINV sequence within genotype 1, the genotype prevalent in Africa and Europe and responsible for most SINV outbreaks (Fig 2). Moreover, the sequence from Burkina Faso fell within a cluster of sequences, mostly from central Africa, that has been shown to be at the origin of the SINV genotypes that have colonized Europe [16]. A similar tree topology was obtained using only the E2 sequence (S1 Fig.).

To estimate the infection rate in mosquito populations, we screened the 59 pools of *Cx. quinquefasciatus* (1 629 mosquitoes in total) for SINV nucleic acids using RT-qPCR, a technique considered more sensitive than metagenomics.

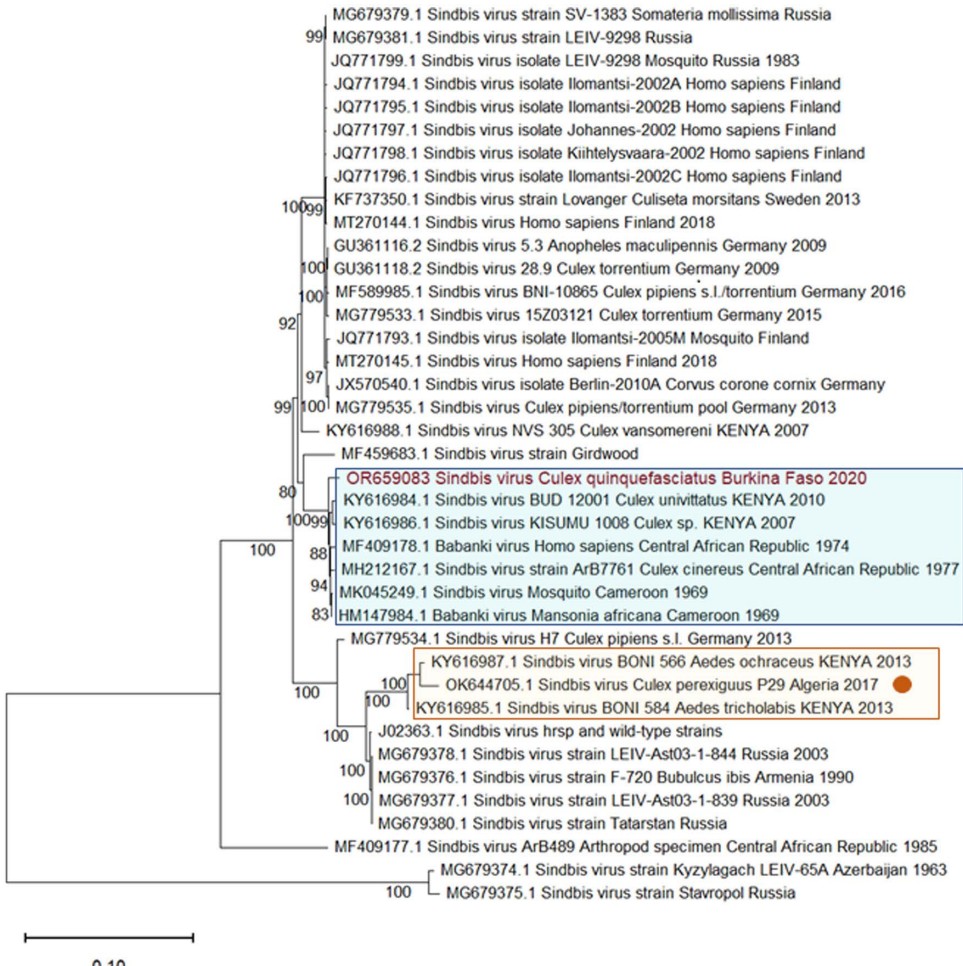

**Fig 2. Maximum-likelihood phylogenetic tree of nucleotide sequences from SINV genotype I.** Sequences are concatenated open reading frames (NS and S ORFs). The name of the sequence from Burkina Faso is shown in red. The sequence from Algeria, from a previous study [32], appears with an orange circle after the name. Two clusters appear underlined with colored rectangles: the cluster containing the sequence from Burkina Faso (blue rectangle) and the cluster containing the sequence from Algeria (orange rectangle). Two sequences from genotype IV were used to root the tree (MG679374 and MG679375). The scale bar represents the number of nucleotide substitutions per site. GenBank accession numbers are indicated on the branch names.

Although *Cx. quinquefasciatus* has not been shown to be infected by SINV in nature, we restricted the screening to this mosquito species because the main SINV vectors usually belong to the genus *Culex* [7,16,32]. Two pools were found to be positive for SINV nucleic acids (Ct values = 22 and 33). These two pools had been used to generate the SINV-positive library (Table S3). To confirm that the detection of SINV nucleic acids was from a viable virus, we attempted to isolate the virus in cell culture from the two SINV-positive pools. Cytopathic effects were observed in the cultures inoculated with the pool with the highest virus concentration as shown by RT-qPCR. SINV isolation was confirmed by PCR (Ct value in culture supernatant = 7). We then estimated the SINV infection rate among *Cx. quinquefasciatus* females in Rural 1 zone in 2020. The estimation assumed that the pool that resulted in cell culture infection was the only one containing SINV-infected mosquitoes. The maximum likelihood estimate of the infection rate was 0.57% (95% confidence intervals = 0.24%/4.26%).

## Discussion

Our working hypothesis was that shotgun metagenomics could provide the full diversity of arboviruses transmitted by mosquitoes in a given region, facilitating follow-up studies with targeted diagnostics. Our metagenomics-based survey allowed the detection of SINV, a zoonotic arbovirus, in a recent collection of mosquito vectors. To our knowledge, SINV has not been detected in Burkina Faso before. Detection of SINV was validated by virus isolation and PCR from non-blood-engorged females. Previously, a PCR-based survey on the same samples failed to detect six arboviruses potentially present in the country (DENV, ZIKV, CHIKV, WNV and Usutu virus) [20,21], which we confirmed with this metagenomic survey. Comparison of the surveys using either PCR or metagenomics suggests that screening mosquito collections with viral metagenomics could allow a comprehensive exploration of the viral diversity in regions with a high viral diversity, as shown by others [51,52]. Nevertheless, metagenomics is still financially and technically demanding [53]. These limitations should be weighed against the benefits of identifying the circulation of a new pathogen for case diagnosis and patient management. Here, we could not determine the potential benefits because the prevalence of SINV infections in humans has not yet been studied in Burkina Faso. This is a major limitation of our study. In fact, studies on the prevalence of SINV infections in humans in African countries are scarce [31,54]. The only recent report found an incidence of 12.7% in patients with acute febrile illness of unknown cause found in South Africa in 2019 and 2020 [31]. Studies on the prevalence (*e.g.*, serosurveys) and symptomatology of SINV infections in humans are needed to assess the potential health impact of SINV in Burkina Faso. These studies may also facilitate the estimation of the benefits of the metagenomics-based survey.

This first robust detection of SINV in West Africa could be the result of either a recent introduction into the region, or an undetected endemic situation. This question also applies to the recent detection of SINV in Northwest Africa [32]. Undetected endemicity in Burkina Faso seems plausible given the limited number of studies on SINV and its large geographic distribution in Africa [15]. However, a recent spread of the virus cannot be ruled out because SINV spread has been observed both within and between continents [16,55]. Longitudinal surveys of SINV prevalence and genetics, including retrospective analyses of sample collections, could help to determine whether the same genotype has been present over years, thus supporting endemicity [56].

The phylogenetic analysis provided two interesting results regarding the geographic distribution of SINV. First, the genomic sequence from Burkina Faso fell within the cluster of sequences from central and eastern Africa, which is supposedly at the origin of the European strains [16]. Thus, West Africa should also be considered among the potential geographic sources of the European strains. However, other scenarios could have taken place. For example, West Africa could simply be a recipient of strains from Central Africa, without being involved in the spread of SINV to Europe. More sequences from West Africa are needed to address this question through a robust phylodynamic analysis. Second, the sequence from Burkina Faso fell in a different cluster than that of a sequence, also of genotype 1, recently identified in northwest Africa [32] (Fig 2). Thus, if SINV has recently spread in West and Northwest Africa, then two distinct dispersals have occurred because the strains involved are different. In addition, our results further support that different SINV strains are circulating in different African regions, calling for studies of SINV diversity and geographic range in Africa.

SINV was detected in a rural area, near the second largest city in Burkina Faso, and only in 2020. Detection in a rural area is consistent with the enzootic cycle of SINV mainly involving wild birds and ornithophilic *Culex* mosquitoes. In this study, the limited number of *Cx. quinquefasciatus* mosquitoes per year and zone (116 females on average) prevented a robust quantitative analysis of SINV prevalence over years. Despite this limitation, two results support the need for future assessment of the potential epidemiologic risk of SINV in Burkina Faso. First, we detected potential SINV infection of *Cx. quinquefasciatus* in the wild for the first time. Previously, SINV infection of *Cx. quinquefasciatus* had only been observed in laboratory experiments with a long-established colony. This mosquito is a major vector of several mosquito-borne pathogens, including viruses (*e.g.*, WNV and St. Louis encephalitis virus) and filarial nematodes in subtropical and tropical areas worldwide. In contrast with *Cx. univitattus*, considered the major vector of SINV in most African countries and

usually found in rural environments [7,16], *Cx. quinquefasciatus* is abundant in both urban and rural areas in Burkina Faso [34]. Moreover, although mainly ornithophilic, this mosquito readily feeds on humans [57]. SINV has been isolated from the saliva of experimentally-infected females, strongly suggesting that *Cx. quinquefasciatus* could be a SINV vector [58]. Hence, if its role as a SINV vector in nature is confirmed, *Cx. quinquefasciatus* has several characteristics that should increase the risk of spillover from an enzootic cycle to humans. The second result related to epidemiologic risk is the prevalence of SINV in *Cx. quinquefasciatus* populations. The infection rate in the area and year of detection was often within the same order of magnitude as estimates from mosquito populations in Europe, including infection rates during outbreaks [59–61]. However, the limited extent of the number, geographic range and time period of SINV-positive pools did not allow for an accurate estimate, as shown by the confidence intervals. This situation does not allow robust conclusions to be drawn on SINV prevalence. Larger mosquito sampling and screening campaigns are needed to accurately assess the prevalence of SINV in mosquitoes in the region.

Burkina Faso, like many African countries, harbors numerous and taxonomically-diverse arboviruses that can affect human health [12,22,23,34]. Information on this diversity and its prevalence can help define needs in surveillance and diagnostic. Moreover, this knowledge can help to understand the spread of viruses within and between countries. In this context, the interest of our results is twofold. First, our results support the interest of shotgun metagenomics in surveys for arboviruses in a country with a high viral diversity. Second, the detection of SINV paves the way for studies on the prevalence of SINV infection in humans in Burkina Faso and other countries in the West African region.

## Supporting information

**S1 Table. Metadata of the pools used in the study.** For each pool, Table S1 provides the mosquito species, site, year, number of mosquitoes, associated library and replicate (replicate: pools of the same site and date).
(CSV)

**S1 Data. Supplementary Information.** This file contains S2 Table, S1 Script, S3 Table and S1 Fig. The titles of each item are: S2 Table. Primers used to amplify and sequence gaps in the SINV genome. S1 Script. R code used to estimate the SINV infection rate. S3 Table. Metadata of the pools from the Rural 1 zone obtained in 2020, including information on the pool that led to SINV isolation in cell culture (SINV_positive = 1). S1 Fig. Maximum-likelihood phylogeny based on the full ORF sequence of the E2 gene. The sequence from Burkina Faso is indicated with a black square. The scale bar represents the number of nucleotide substitutions per site. GenBank accession numbers are indicated on the branch names. The phylogenetic tree was generated following the procedure described in the Materials and Methods section of the main text.
(PDF)

## Author contributions

**Conceptualization:** Didier P. Alexandre Kaboré, Philippe Van de Perre, Thierry Baldet, Roch K. Dabiré, Serafin Gutierrez.

**Data curation:** Didier P. Alexandre Kaboré.

**Formal analysis:** Didier P. Alexandre Kaboré, Patricia Gil.

**Funding acquisition:** Philippe Van de Perre, Thierry Baldet, Roch K. Dabiré, Serafin Gutierrez.

**Investigation:** Didier P. Alexandre Kaboré, Dieudonné Diloma Soma, Simon P. Sawadogo, Georges Anicet Ouédraogo, Patricia Gil, Serafin Gutierrez.

**Methodology:** Didier P. Alexandre Kaboré, Roch K. Dabiré, Serafin Gutierrez.

**Project administration:** Patricia Gil.

**Software:** Antoni Exbrayat, Floriant Charriat.

**Supervision:** Roch K. Dabiré, Patricia Gil, Serafin Gutierrez.

**Validation:** Edouard Tuaillon.

**Visualization:** Côme Morel.

**Writing – original draft:** Didier P. Alexandre Kaboré.

**Writing – review & editing:** Didier P. Alexandre Kaboré, Antoni Exbrayat, Floriant Charriat, Dieudonné Diloma Soma, Simon P. Sawadogo, Georges Anicet Ouédraogo, Edouard Tuaillon, Philippe Van de Perre, Thierry Baldet, Côme Morel, Roch K. Dabiré, Patricia Gil, Serafin Gutierrez.

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
