## [Decision Letter · Decision Letter 0]

25 Nov 2024

Dear Dr. Gutierrez,

Thank you for submitting your manuscript to PLOS ONE. After careful consideration, we feel that it has merit but does not fully meet PLOS ONE’s publication criteria as it currently stands. Therefore, we invite you to submit a revised version of the manuscript that addresses the points raised during the review process.

There were some minor comments and clarifications requested by reviewer 1 which would improve this manuscript.

We look forward to receiving your revised manuscript.

Kind regards,

Kelli L. Barr, Ph.D.

Academic Editor

PLOS ONE

Journal Requirements:

“This work was supported by the Montpellier University of Excellence programme, 2018 (ArboSud project; authors receiving the grant: P.V.T., R.D., T.B., S.G.). D.A.P.K. was the recipient of a funding by European Union’s Horizon 2020 research and innovation programme (Infravec2 project, grant agreement 731060). The funders had no role in study design, data collection and analysis, decision to publish, or preparation of the manuscript.”

“This work is part of the ArboSud project funded by the 2018 call of the Montpellier University of Excellence (MUSE) program. D. Kabore acknowledges funding from the European Union’s Horizon 2020 research and innovation program (Infravec2 project, grant agreement 731060).”

“This work was supported by the Montpellier University of Excellence programme, 2018 (ArboSud project; authors receiving the grant: P.V.T., R.D., T.B., S.G.). D.A.P.K. was the recipient of a funding by European Union’s Horizon 2020 research and innovation programme (Infravec2 project, grant agreement 731060). The funders had no role in study design, data collection and analysis, decision to publish, or preparation of the manuscript.”

“The authors declare no competing interests”

5. Thank you for uploading your study's underlying data set. Unfortunately, the repository you have noted in your Data Availability statement does not qualify as an acceptable data repository according to PLOS's standards.

6. In the online submission form, you indicated that your data will be submitted to a repository upon acceptance. We strongly recommend all authors deposit their data before acceptance, as the process can be lengthy and hold up publication timelines. Please note that, though access restrictions are acceptable now, your entire minimal dataset will need to be made freely accessible if your manuscript is accepted for publication. This policy applies to all data except where public deposition would breach compliance with the protocol approved by your research ethics board. If you are unable to adhere to our open data policy, please kindly revise your statement to explain your reasoning and we will seek the editor's input on an exemption.

7. We note that Figure 7 in your submission contain map images which may be copyrighted. All PLOS content is published under the Creative Commons Attribution License (CC BY 4.0), which means that the manuscript, images, and Supporting Information files will be freely available online, and any third party is permitted to access, download, copy, distribute, and use these materials in any way, even commercially, with proper attribution. For these reasons, we cannot publish previously copyrighted maps or satellite images created using proprietary data, such as Google software (Google Maps, Street View, and Earth). For more information, see our copyright guidelines: http://journals.plos.org/plosone/s/licenses-and-copyright.

a. You may seek permission from the original copyright holder of Figure 7 to publish the content specifically under the CC BY 4.0 license. 

Reviewers' comments:

Reviewer's Responses to Questions

**Comments to the Author**

1. Is the manuscript technically sound, and do the data support the conclusions?

Reviewer #1: Yes

Reviewer #2: Yes

2. Has the statistical analysis been performed appropriately and rigorously?

Reviewer #1: Yes

Reviewer #2: Yes

3. Have the authors made all data underlying the findings in their manuscript fully available?

Reviewer #1: Yes

Reviewer #2: Yes

4. Is the manuscript presented in an intelligible fashion and written in standard English?

Reviewer #1: Yes

Reviewer #2: Yes

Reviewer #1: General comments

This work looks like the first of its kind in Western Africa. In general, and with the first detection of SINV in Burkina Faso this manuscript is a valuable contribution to increasing knowledge on the prevalence of viruses in West Africa. As this knowledge is still very limited in many African regions, every report including genetic information by sequencing can be useful to close this gap.

In anticipation of the result and discussion section, the authors should not underestimate the value of this report. The authors discuss the use of metagenomic screening surveys, such as the one presented in the study, as an alternative to targeted diagnostics for arbovirus diversity in mosquitoes. However, it would be more appropriate to frame this work as a supplementary or pilot study to assess viral prevalence in this region. With this and after knowing that the virus is prevalent in the vector, follow-up studies (in vectors or humans) with a targeted detection approach can serve as a more specific and less expensive detection approach.

The paper is well-structured. The English language level is good but some words and sentences need editing.

Specific comments

In the author's summary of the work, in lines 51-52, it was stated that this is the first time SINV has been found in Culex quinquefasciatus. Is this so in Burkina Faso or worldwide?

Introduction

1. In lines 81-83, the authors should check this statement and consider these questions: Are arboviruses specific to vectors in terms of species? For the disease to be endemic, it means such vectors should be found in that area, are the vectors globally distributed? Or geographically oriented? How then did they result in outbreaks? Or is this statement a general observation for the spread of any virus not just arboviruses?

2. In line 88, there is a grammatical error. The use of no specific ...... and unavailable. If no treatments are available, it is either "specific treatments... are unavailable" or "no specific treatments... are available." In that same line, the statement is a bit confusing. What do the authors mean by unnecessary diagnostic testing? Are arboviral infections disguisable by signs and symptoms? Or do you mean about other causative organisms apart from arboviruses?

3. The second statement in line 104, can be backed with literature? It would give some knowledge of human behavior's effect on arboviral diversity.

Methods

Regarding the general methodology, the study design and implementation were conducted appropriately. However, there is some need for clarification in the methods and results section.

1. The first statement for sampling sites is great and simple but it will be a bit inconvenient to go and read that paper on mosquito diversity before getting the rationale for the choice of sampling sites. I think the authors can give a brief reason then the citation can be attached if further reading is needed.

2. Three trapping methods were used in this work, were all methods used at each of the different sites? How does this affect the sampling in terms of the number of catches? There is no description of the number of mosquitoes each caught.

3. One major point is the pooling strategy of mosquitoes and the resulting number of libraries that were prepared. According to Table 1, sometimes 3 libraries were prepared from 3 pools, other times 3 libraries were prepared from 5 to 7 pools. The strategy behind this is not clear from the methodology or result section.

Results/Discussion

1. In addition to that, in the result section (line 258) it was stated that the same number of libraries per mosquito species was prepared, but very different numbers of mosquitos were pooled. This results in a different sequencing depth per single mosquito. To achieve a more comparable screening between the pools the same sequencing depth should be targeted across all samples.

2. In line 261, the number of reads per library is given (min/max = 0.8/7.65 million reads per library). The sequencing depth per library varies significantly, is this due to the different numbers of pools per library? A clearer explanation or comment on that would be needed.

3. In line 284 authors state that they only tested Cx. quinquefasciatus mosquitoes in RT-qPCR. For a more comprehensive result, it would have been interesting to also test the other pools of Aedes mosquitoes for SINV. Even reports of negative results would support the thesis that this mosquito species is not the typical vector.

4. Regarding the phylogenetic reconstruction, the general methodology was appropriate. However, the phylogenetic tree in this work is based on concatenated NS and S ORFs. The six SINV genotypes (G1-G6) are usually determined based on E2 gene phylogenies only even when they were first described (Lundström JO, Pfeffer M. 2010. Phylogeographic structure and evolutionary history of Sindbis virus. Vector Borne Zoonotic Dis 10:889–907. doi: 10.1089/vbz.2009.0069). Would a calculation based on the E2 gene change the outcome of phylogenetic reconstruction and cluster the strains differently? This should be tested before considering all the following comments on the phylogeny.

5. In lines 294 – 301 and the corresponding phylogenetic tree, only the cluster of the Burkina Faso strains with other African strains as well as another cluster including the strain from Algeria are described in detail and highlighted. The relation to other strains including the European sequences is not described and also not discussed later. Only in line 341, do the authors hypothesize different genotypes involved in the spread in North and Northwest Africa as well as different scenarios regarding the spread to Europe. However, according to the authors, this phylogeny involves only SINV Genotype 1 sequences. This does not match with the previous result presentation, where SINV sequences were identified as genotype 1. If the authors assume the presence of another SINV genotype (2 – 6), then representative strains should be included in the phylogenetic reconstruction. In case this only refers to the difference in the sequences of Burkina Faso and Algeria strain genomes then this should be explained more precisely because the phylogeny only includes Genotype 1 sequences and show the clusters within this genotype 1.

6. Did the authors also check for major differences in the amino acid sequences of the different ORFs, for example, in a multiple alignment with all the strains that are included in the phylogeny? This can sometimes show patterns of amino acid changes or deletions for a set of strains. It would be interesting to see if there is a notable difference between the sequences from Europe (causing sporadic outbreaks) to those from Burkina Faso and/or African strains in general.

Reviewer #2: I joined the review of the article after the first round of revision, and perhaps that is why the manuscript left the most favorable impression. The authors clearly presented the research methods and results, carefully and self-critically analyzed the data, outlined the limitations of their study, and discussed the significance of their results. It was a stroke of luck to find Sindbis virus in mosquitoes collected in West Africa without any information about an ongoing outbreak. The prevalence estimate in the paper highlights how difficult it is to make such a finding. An important advantage of the work is that the authors not only conducted a metagenomic search, PCR screening, and whole-genome sequencing but also isolated the virus in cell culture. I have no doubts about the importance of the find, and I am grateful to the authors for such a careful and thorough description of the work done.

Then I went through the comments of the first-round reviewers and the authors' responses to them, step by step. And only after this, it became clear that the authors had done a lot of work revising the manuscript. In my opinion, after this conducted revision, the manuscript can be accepted for publication.

There are only 2 small comments, the answers to which do not require a separate round of review:

Lines 154-155. "August/September" means "from August to September" or "only in August and only in September"? Same for "June/October". Please rephrase this text so there are no misunderstandings.

General comment: Based on the previous work (Mosquito (Diptera: Culicidae) populations in contrasting areas of the western regions of Burkina Faso: species diversity, abundance and their implications for pathogen transmission), the authors collected numerous mosquitoes of different species, but PCR screening for Sindbis virus was undertaken only for Cx. quinquefasciatus. It might make sense to extend PCR screening to other mosquito species/genera from Rural 1 zone 2020. It is possible that other species of collected mosquito may also be involved in the circulation of SINV.

**Do you want your identity to be public for this peer review?** For information about this choice, including consent withdrawal, please see our Privacy Policy

Reviewer #1: No

Reviewer #2: **Yes: ** Marat Makenov

---

## [Author Response · Author response to Decision Letter 1]

22 Jan 2025

Dear editor and reviewers,

Please find below a point-by-point answer to the reviewers’ comments. Moreover, the answers to editorial comments can be found below the answer to the reviewers. Finally, we have added a piece of information that could be relevant for the review. This information is the final decision letter from PLoS Neglected Tropical Diseases (see file final decision PLoS Neg Dis in the submitted files). In this letter, one of the reviewers recommends publication.

We thank the reviewers for the improvements in the manuscript.

Best regards,

Serafín Gutiérrez

Reviewer #1: General comments

This work looks like the first of its kind in Western Africa. In general, and with the first detection of SINV in Burkina Faso this manuscript is a valuable contribution to increasing knowledge on the prevalence of viruses in West Africa. As this knowledge is still very limited in many African regions, every report including genetic information by sequencing can be useful to close this gap.

In anticipation of the result and discussion section, the authors should not underestimate the value of this report. The authors discuss the use of metagenomic screening surveys, such as the one presented in the study, as an alternative to targeted diagnostics for arbovirus diversity in mosquitoes. However, it would be more appropriate to frame this work as a supplementary or pilot study to assess viral prevalence in this region. With this and after knowing that the virus is prevalent in the vector, follow-up studies (in vectors or humans) with a targeted detection approach can serve as a more specific and less expensive detection approach.

Following the reviewer’s comment, we have modified the Discussion section as follows:

“Our working hypothesis was that shotgun metagenomics could provide the full diversity of arboviruses transmitted by mosquitoes in a given region, facilitating follow-up studies with targeted diagnostics” (lines 310-312).

The paper is well-structured. The English language level is good but some words and sentences need editing.

Specific comments

1. In the author's summary of the work, in lines 51-52, it was stated that this is the first time SINV has been found in Culex quinquefasciatus. Is this so in Burkina Faso or worldwide?

It is worldwide. This is why the sentence does not include any indication on the country.

Introduction

2. In lines 81-83, the authors should check this statement and consider these questions: Are arboviruses specific to vectors in terms of species? For the disease to be endemic, it means such vectors should be found in that area, are the vectors globally distributed? Or geographically oriented? How then did they result in outbreaks? Or is this statement a general observation for the spread of any virus not just arboviruses?

We understand that reviewer’s comment is due to the fact that arboviruses require populations of mosquito vectors to establish in a given region. This requirement does not apply to viruses that are not transmitted by mosquitoes. The text has been modified to include this information:

“Many of these viruses have periodically spread over long distances and colonized new areas in which populations of mosquito vectors exist” (lines 79-80).

3. In line 88, there is a grammatical error. The use of no specific ...... and unavailable. If no treatments are available, it is either "specific treatments... are unavailable" or "no specific treatments... are available." In that same line, the statement is a bit confusing. What do the authors mean by unnecessary diagnostic testing? Are arboviral infections disguisable by signs and symptoms? Or do you mean about other causative organisms apart from arboviruses?

We thank the reviewer for spotting the grammatical error. The current version follows the first option proposed by the reviewer (“specific treatments… are unavailable”).

In the same line, we agree that the term “unnecessary” is not appropriate here. The term in the current version is “unspecific”. We believe this term better describes the tests for different pathogens carried out to investigate a case due to an arbovirus not known to circulate in the region. Tests against different organisms are usually required because symptoms of many arboviral infections are often not distinguishable of those provoked by other causative organisms.

4. The second statement in line 104, can be backed with literature? It would give some knowledge of human behavior's effect on arboviral diversity.

Human activity has an impact on the taxonomic diversity of mosquito vectors and, thus, on the viral diversity. For example, several mosquito vectors, like Aedes albopictus, are usually more abundant in urbanized areas whereas others, like Culex pipiens, can be found in greater numbers in rural areas. We have added a reference to back the statement (reference 26: Mosquito communities and disease risk influenced by land use change and seasonality in the Australian tropics).

Methods

Regarding the general methodology, the study design and implementation were conducted appropriately. However, there is some need for clarification in the methods and results section.

5. The first statement for sampling sites is great and simple but it will be a bit inconvenient to go and read that paper on mosquito diversity before getting the rationale for the choice of sampling sites. I think the authors can give a brief reason then the citation can be attached if further reading is needed.

A short description of the logics behind the choice of sampling sites is now provided in lines 133-134: “The choice of the sampling sites aimed at exploring the viral diversity in different geographical regions and human activity levels”.

6. Three trapping methods were used in this work, were all methods used at each of the different sites? How does this affect the sampling in terms of the number of catches? There is no description of the number of mosquitoes each caught.

The use of all methods in all sites is now clearly stated (line 159). Moreover, we now indicate that the description of mosquito catches per sampling method is available in our previous article on the entomological survey (line 162).

7. One major point is the pooling strategy of mosquitoes and the resulting number of libraries that were prepared. According to Table 1, sometimes 3 libraries were prepared from 3 pools, other times 3 libraries were prepared from 5 to 7 pools. The strategy behind this is not clear from the methodology or result section.

In addition to that, in the result section (line 258) it was stated that the same number of libraries per mosquito species was prepared, but very different numbers of mosquitos were pooled. This results in a different sequencing depth per single mosquito. To achieve a more comparable screening between the pools the same sequencing depth should be targeted across all samples.

The strategy for library production aimed at generating similar numbers of libraries for both mosquito species in each site and period. Moreover, our strategy was to screen all individuals from the two mosquito species. The pooling strategy in library preparation is now more detailed (lines 191-193).

Catches of each species differed depending on sites and periods, and more individuals of Culex quinquefasciatus were captured in total (273 individuals more). This situation led to differences in the number of pools or mosquitoes per library. This disparity between species and sampling points is commonplace in mosquito sampling.

Despite this disparity, the pooling design generated a similar number of libraries for the two mosquito species per site and period (in total, 32 and 34 libraries for each mosquito species). Moreover, the range of mosquitoes per library was similar between mosquito species (min/max per library = 25/62 and 24/55 for Cx. quinquefasciatus and Ae. aegypti respectively). Finally, the mean number of viral reads per individual mosquito did not greatly differ between mosquito species (657 and 804 for Cx. quinquefasciatus and Ae. aegypti respectively). This last piece of information is now provided in the manuscript (lines 265-267).

8. In line 261, the number of reads per library is given (min/max = 0.8/7.65 million reads per library). The sequencing depth per library varies significantly, is this due to the different numbers of pools per library? A clearer explanation or comment on that would be needed.

The sequencing depth did not positively correlate with mosquito number (y = -57584x + 6425274, R2 = 0.295). In our hands, the disparity in the reads per library is a common observation. This disparity probably comes from two main sources. The first one is the difficulty to mix a large number of libraries (66 libraries here) at the same final concentration without diluting in excess the final pool. The second source is the inherent random nature of the Illumina technology. We routinely quantify our libraries using real-time PCR. We also perform a low-depth sequencing of our libraries (2 million reads with a nano cell in a MiSeq instrument) to analyze potential differences in reads per library.

9. In line 284 authors state that they only tested Cx. quinquefasciatus mosquitoes in RT-qPCR. For a more comprehensive result, it would have been interesting to also test the other pools of Aedes mosquitoes for SINV. Even reports of negative results would support the thesis that this mosquito species is not the typical vector.

We agree that negative results are also informative. However, they are informative if sample size is enough large to limit the chances of false negatives. Unluckily, here, two factors lead to a lack of robustness in the proposed analysis. First, SINV prevalence was very low in Culex mosquitoes. This situation implies that a large number of Aedes females has to be tested to ascertain their vector role. Secondly, the number of Aedes mosquitoes was low in the site and period in which the positive pool was found (51 females distributed in 2 pools). Testing two pools cannot provide a robust answer on the potential absence of vector role of Aedes aegypti. Thus, we prefer not to carry out the test since it cannot provide a robust answer.

10. Regarding the phylogenetic reconstruction, the general methodology was appropriate. However, the phylogenetic tree in this work is based on concatenated NS and S ORFs. The six SINV genotypes (G1-G6) are usually determined based on E2 gene phylogenies only even when they were first described (Lundström JO, Pfeffer M. 2010. Phylogeographic structure and evolutionary history of Sindbis virus. Vector Borne Zoonotic Dis 10:889–907. doi: 10.1089/vbz.2009.0069). Would a calculation based on the E2 gene change the outcome of phylogenetic reconstruction and cluster the strains differently? This should be tested before considering all the following comments on the phylogeny.

We agree that authors often use a partial sequence of E2 gene to generate SINV phylogenies, probably due to historical and practical reasons (i.e. availability of primers). We have generated a phylogeny on a concatenation of all ORFs because this approach has been shown to be more precise to discriminate within-clade structure (Ling J, et al. 2019. Introduction and dispersal of Sindbis virus from central Africa to Europe. J Virol 93:e00620-19. https://doi.org/10.1128/JVI.00620-19.).

Following the reviewer’s comment, we have generated a phylogeny based on the E2 gene. The resulting tree is very similar to the one provided in the manuscript. The tree based on E2 is provided in the supplementary material (Sup. Fig. 1).

5. In lines 294 – 301 and the corresponding phylogenetic tree, only the cluster of the Burkina Faso strains with other African strains as well as another cluster including the strain from Algeria are described in detail and highlighted. The relation to other strains including the European sequences is not described and also not discussed later. Only in line 341, do the authors hypothesize different genotypes involved in the spread in North and Northwest Africa as well as different scenarios regarding the spread to Europe. However, according to the authors, this phylogeny involves only SINV Genotype 1 sequences. This does not match with the previous result presentation, where SINV sequences were identified as genotype 1. If the authors assume the presence of another SINV genotype (2 – 6), then representative strains should be included in the phylogenetic reconstruction. In case this only refers to the difference in the sequences of Burkina Faso and Algeria strain genomes then this should be explained more precisely because the phylogeny only includes Genotype 1 sequences and show the clusters within this genotype 1.

We apologize for the lack of clarity. The term “genotype” was used in the sense used in evolutionary biology (i.e. a genome with a specific sequence) rather than in the sense used in virology (i.e. a group of related genomic sequences). We have modified the text to clarify the term for readers from both fields. The term genotype has been replaced by the term “strain” (lines 347-351).

6. Did the authors also check for major differences in the amino acid sequences of the different ORFs, for example, in a multiple alignment with all the strains that are included in the phylogeny? This can sometimes show patterns of amino acid changes or deletions for a set of strains. It would be interesting to see if there is a notable difference between the sequences from Europe (causing sporadic outbreaks) to those from Burkina Faso and/or African strains in general.

We also find the question interesting. A previous paper has already analyzed this question in depth and including most of the sequences in our phylogenetic tree (Ling J, et al. 2019. Introduction and dispersal of Sindbis virus from central Africa to Europe. J Virol 93:e00620-19. https://doi.org/10.1128/JVI.00620-19.). This work has not found major differences in percent identities between African and European sequences. More precisely, percent identities ranged from 99.1 to 100% for both nucleotide and amino acid sequences. An analysis including our sequence would not deviate from those findings. For the sake of conciseness, we prefer not to include the proposed analysis.

Reviewer #2: I joined the review of the article after the first round of revision, and perhaps that is why the manuscript left the most favorable impression. The authors clearly presented the research methods and results, carefully and self-critically analyzed the data, outlined the limitations of their study, and discussed the significance of their results. It was a stroke of luck to find Sindbis virus in mosquitoes collected in West Africa without any information about an ongoing outbreak. The prevalence estimate in the paper highlights how difficult it is to make such a finding. An important advantage of the work is that the authors not only conducted a metagenomic search, PCR screening, and whole-genome sequencing but also isolated the virus in cell culture. I have no doubts about the importance of the find, and I am grateful to the authors for such a careful and thorough description of the work done.

Then I went through the comments of the first-round reviewers and the authors' responses to them, step by step. And only after this, it became clear that the authors had done a lot of work revising the manuscript. In my opinion, after this conducted revision, the manuscript can be accepted for publication.

There are only 2 small comments, the answers to which do not require a separate round of review:

Lines 154-155. "August/September" means "from August to September" or "only in August and only in September"? Same for "June/October". Please rephrase this text so there are no misunderstandings.

We thank the reviewer for having spotted this clarity problem. The correct syntaxis is “from … to …”. We have rephrased the text accordingly.

General comment: Based on the previous work (Mosquito (Diptera: Culicidae) populations in contrasting areas of the western regions of Burkina Faso: species diversity, abundance and their implications for pathogen transmission), the authors collected numerous mosquitoes of different species, but PCR screening for Sindbis virus was undertaken only for Cx. quinquefasciat

---

## [Decision Letter · Decision Letter 1]

15 Apr 2025

A metagenomics survey of viral diversity in mosquito vectors allows the first detection of Sindbis virus in Burkina Faso

PONE-D-24-41081R1

Dear Dr. Gutierrez,

We’re pleased to inform you that your manuscript has been judged scientifically suitable for publication and will be formally accepted for publication once it meets all outstanding technical requirements.

Please during final proofing make the changes suggested by reviewer 2, which are small enough to be introduced during copyediting.

Kind regards,

Rhys Harold Parry

Academic Editor

PLOS ONE

Additional Editor Comments (optional):

Reviewers' comments:

Reviewer's Responses to Questions

**Comments to the Author**

Reviewer #2: (No Response)

Reviewer #3: (No Response)

2. Is the manuscript technically sound, and do the data support the conclusions?

Reviewer #2: Yes

Reviewer #3: Yes

3. Has the statistical analysis been performed appropriately and rigorously?

Reviewer #2: Yes

Reviewer #3: Yes

4. Have the authors made all data underlying the findings in their manuscript fully available?

Reviewer #2: Yes

Reviewer #3: Yes

5. Is the manuscript presented in an intelligible fashion and written in standard English?

Reviewer #2: Yes

Reviewer #3: Yes

Reviewer #2: "August/September" and "June/October" are still confusing, but that's a minor quibble. At the discretion of the authors

Reviewer #3: I commend the authors for their valuable contribution to arbovirus surveillance in mosquito vectors. Arboviruses in vectors remain significantly understudied, particularly in Africa, where their circulation and potential impact on public health are not well documented. The experiments have been conducted rigorously, and the study provides valuable insights.

That said, I noticed that a comment from Reviewer 2 regarding Lines 157–158 has not been fully addressed. The phrasing “August/September 2019,” “June/October 2020,” and “May/June 2021” remains ambiguous. To enhance clarity, I recommend explicitly stating “from August to September”,“from June to October” and “from May to June “ to prevent any misinterpretation.

Additionally, I recommend including the sampling years in the abstract. Specifying the study period upfront would improve clarity and accessibility, ensuring that readers do not have to search for this information in the methodology section. This would make the abstract more informative and enhance readability.

Otherwise, the manuscript is well-structured and presents significant findings that contribute to the field

**Do you want your identity to be public for this peer review?** For information about this choice, including consent withdrawal, please see our Privacy Policy

Reviewer #2: No

Reviewer #3: No

---

## [Editor Report · Acceptance letter]

PONE-D-24-41081R1

PLOS ONE

Dear Dr. Gutierrez,

I'm pleased to inform you that your manuscript has been deemed suitable for publication in PLOS ONE. Congratulations! Your manuscript is now being handed over to our production team.

Kind regards,

on behalf of

Dr. Rhys Harold Parry

Academic Editor

PLOS ONE